# Graph-Based Electroencephalography Analysis in Tinnitus Therapy

**DOI:** 10.3390/biomedicines12071404

**Published:** 2024-06-25

**Authors:** Muhammad Awais, Khelil Kassoul, Abdelfatteh El Omri, Omar M. Aboumarzouk, Khalid Abdulhadi, Samir Brahim Belhaouari

**Affiliations:** 1Department of Creative Technologies, Air University, Islamabad 44000, Pakistan; muhammadawais95@gmail.com; 2Geneva School of Business Administration, University of Applied Sciences Western Switzerland, HES-SO, 1227 Geneva, Switzerland; 3Surgical Research Section, Department of Surgery, Hamad Medical Corporation, Doha P.O. Box 3050, Qatar; aelomri@hamad.qa (A.E.O.); oaboumarzouk@hamad.qa (O.M.A.); 4Vice President for Medical and Health Sciences Office, QU-Health, Qatar University, Doha P.O. Box 2713, Qatar; 5Department of Clinical Science, College of Medicine, Qatar University, Doha P.O. Box 2713, Qatar; 6School of Medicine, Dentistry and Nursing, The University of Glasgow, Glasgow G12 8QQ, UK; 7Audiology and Balance Unit, Hamad Medical Corporation (HMC), Doha P.O. Box 3050, Qatar; khadi@hamad.qa; 8Division of Information and Computing Technology, College of Science and Engineering, Hamad Bin Khalifa University, Doha P.O. Box 5825, Qatar

**Keywords:** tinnitus dataset, electroencephalography (EEG) signals, preprocessing techniques, feature extraction, Graph Neural Networks (GNNs)

## Abstract

Tinnitus is the perception of sounds like ringing or buzzing in the ears without any external source, varying in intensity and potentially becoming chronic. This study aims to enhance the understanding and treatment of tinnitus by analyzing a dataset related to tinnitus therapy, focusing on electroencephalography (EEG) signals from patients undergoing treatment. The objectives of the study include applying various preprocessing techniques to ensure data quality, such as noise elimination and standardization of sampling rates, and extracting essential features from EEG signals, including power spectral density and statistical measures. The novelty of this research lies in its innovative approach to representing different channels of EEG signals as new graph network representations without losing any information. This transformation allows for the use of Graph Neural Networks (GNNs), specifically Graph Convolutional Networks (GCNs) combined with Long Short-Term Memory (LSTM) networks, to model intricate relationships and temporal dependencies within the EEG data. This method enables a comprehensive analysis of the complex interactions between EEG channels. The study reports an impressive accuracy rate of 99.41%, demonstrating the potential of this novel approach. By integrating graph representation and deep learning, this research introduces a new methodology for analyzing tinnitus therapy data, aiming to contribute to more effective treatment strategies for tinnitus sufferers.

## 1. Introduction

Tinnitus is a condition where individuals perceive sound even when there is no external source of sound [1]. This sensation is often described as hearing ringing, buzzing, hissing, or clicking sounds in the ears [2]. Tinnitus can manifest in one or both ears and can vary in terms of its intensity and frequency [3]. The prevalence of tinnitus in the population can vary depending on the specific study and group of people being examined [4,5]. Estimates suggest that approximately 10–15% of adults experience chronic tinnitus [6], whereas as many as 30% may encounter occasional or transient tinnitus [7]. Tinnitus is not limited to a particular age group [8], but it tends to be more commonly reported among older adults [9].

Tinnitus can stem from various underlying causes [10], including exposure to loud noises, age-related hearing loss, ear infections, specific medications, blockage by earwax, head and neck injuries, and other medical conditions [11]. The impact of tinnitus on individuals can range from mild annoyance to significant distress [12,13], affecting their overall quality of life, sleep patterns [14], ability to concentrate, and emotional well-being.

### 1.1. Tinnitus Treatment and Management Strategies

Tinnitus management encompasses various strategies to alleviate symptoms [7], including sound therapy, cognitive behavioral therapy, relaxation techniques, and the use of hearing aids or masking devices [15,16]. The approach to treating and managing tinnitus varies based on the underlying cause and individual factors [17]. While there is currently no known cure for tinnitus, numerous approaches exist to alleviate symptoms and enhance quality of life [18]. Figure 1 presents the common strategies employed to manage tinnitus across all age groups.

### 1.2. Effectiveness of Various Therapies

Researchers have examined the effectiveness of five different therapies [18] for reducing tinnitus perception in patients with refractory and chronic tinnitus. These therapies covered the various elements outlined in Table 1.

To assess therapeutic effectiveness, researchers analyzed patients’ electroencephalographic (EEG) activity [19] and measured brainwave patterns associated with tinnitus perception [20]. By comparing EEG data before and after therapy sessions [21], researchers have aimed to determine the impact of each therapy on reducing tinnitus-related brainwave patterns and improving patients’ subjective experiences [22]. This analysis provides objective measurements to evaluate the potential effectiveness of these therapies in individuals with refractory and chronic tinnitus.

**Table 1 biomedicines-12-01404-t001:** Comprehensive aspects encompassed by therapeutic approaches.

Therapy	Aspect
Relaxing Music	This therapy involves exposing patients to soothing music aimed at promoting relaxation and reducing stress levels. The goal was to create a more favorable auditory environment to minimize tinnitus perception [23].
Tinnitus Retraining Therapy (TRT)	Focusing on habituation and desensitization techniques, TRT aims to reclassify tinnitus from bothersome to neutral. It utilizes sound therapy and counseling to facilitate habituation [24].
Therapy for Enriched Acoustic Environment (TEAE).	TEAE sought to enrich the acoustic environment by introducing various background sounds to make tinnitus less noticeable and bothersome [25].
Binaural Beats Therapy (TBB)	TBB involves listening to auditory stimuli with slightly different frequencies in each ear to induce relaxation and reduce tinnitus perception [26].
Auditory Discrimination Therapy (ADT)	ADT aims to enhance auditory processing by training patients to discriminate between different sounds, including tinnitus [18].

In recent years, there has been an increasing focus on harnessing deep learning methodologies for the identification of tinnitus through the analysis of EEG signals [27] recorded during therapy sessions. In their study [27], the authors aimed to establish an objective methodology for assessing changes in attentional processes among patients with tinnitus undergoing auditory discrimination therapy (ADT) using EEG signals. Chronic and refractory tinnitus have been associated with neuronal over-synchronization, and sound-based therapies have emerged as potential treatment approaches. However, the effect of ADT on attentional processes remains poorly understood. This study utilized event-related (de-)synchronization (ERD/ERS) responses to map synchronization levels related to auditory recognition events. Deep representations of the scalograms were then extracted using a pre-trained Convolutional Neural Network (CNN) architecture (MobileNet v2). These deep-spectrum features were analyzed to assess changes in attention and memory performance within the study datasets. The results provide robust evidence supporting the feasibility of ADT as a tinnitus treatment, potentially due to attentional redirection. This research contributes to a better understanding of the effects of ADT on patients with tinnitus and introduces an objective measurement approach using EEG signals to monitor attentional changes throughout the therapy process.

The author [28] presents a methodology designed to tackle the challenges posed by tinnitus, a persistent condition characterized by the perception of sound without any external source. The diversity of symptom profiles in patients with tinnitus presents a major hurdle for the development of effective treatments. In the absence of universal treatment, patients often seek a promising approach to alleviate tinnitus, even without empirical support. Furthermore, the lack of objective measures for assessing individual symptoms hinders comprehension and management of the disorder. To address these challenges, the author investigated the use of EEG data reflecting brain activity for classifying tinnitus using a deep neural network. The study encompasses the analysis of 16,780 raw EEG samples from 42 subjects, including both tinnitus patients and a control group, each with a one-second duration. Four distinct automated preprocessing techniques, including noise reduction and various sampling strategies, were compared. Following preprocessing, a neural network was trained to classify whether a given sample corresponded to a patient with tinnitus or to the control group. The findings demonstrate a maximum accuracy of 75.6% in the test set when applying noise reduction and down-sampling during preprocessing. These results underscore the potential of deep learning methods for detecting complex EEG patterns associated with tinnitus, which are challenging for humans to discern. The proposed methodology shows the effectiveness of leveraging deep learning techniques and EEG data for tinnitus detection and classification. By advancing our understanding of tinnitus through objective measurements and machine learning, this study offers valuable insights into the development of improved diagnostic and therapeutic strategies.

The authors of this study [29] set out to explore how modulated acoustic stimulation affects the brain network dynamics of individuals suffering from chronic tinnitus. Their research involved the design of a paradigm for capturing EEGs from patients with tinnitus undergoing consecutive neuromodulation therapy involving acoustic stimulation for up to 75 days. Tinnitus severity and treatment efficacy were assessed using the tinnitus handicap inventory (THI), while EEG recordings documented brain activities at two-week intervals. The authors conducted an EEG-based coherence analysis to investigate whether changes in the brain network were aligned with the observed clinical outcomes during the 75-day acoustic treatment period. Subsequently, a correlation analysis was performed to uncover potential relationships between network properties and alterations in tinnitus handicap inventory scores. The results revealed significant weakening of the EEG network following extended periodic acoustic stimulation treatment. Notably, strong correlations were identified between changes in tinnitus handicap inventory scores, treatment efficacy, and variations in the brain network properties. These findings suggest that long-term acoustic stimulation neuromodulation interventions are promising for improving the rehabilitation of patients with chronic tinnitus.

This study aimed to introduce an automated and impartial method to assess both the presence and severity of tinnitus in patients undergoing therapy. By harnessing the capabilities of deep learning algorithms known for their proficiency in extracting intricate patterns and representations from complex datasets, we hold high expectations for achieving precise tinnitus detection.

In the scope of this study, we focused on the analysis of a tinnitus dataset, specifically comprising therapy data [18]. More specifically, we investigated EEG signals originating from patients engaged in tinnitus therapy. To prepare EEG data for analysis, we applied a range of preprocessing techniques. These methods encompass the removal of noise artifacts [30], ensuring uniform sampling rates, and normalizing the data to enhance comparability across various recordings [31]. In addition, we performed feature extraction to encapsulate the pertinent attributes of the EEG signals. These features commonly encompass time-domain or frequency-domain characteristics such as power spectral density [32], spectral entropy, and statistical measures [33]. These features provide the data with meaningful representations.

Following the preparation of EEG signal features, we embarked on an innovative approach: converting the signals into graphical representations [34]. Each cluster of nodes was linked by edges to form a graph structure that mirrored the data. This graph framework enabled us to harness Graph Neural Networks (GNNs), particularly the Graph Convolutional Network (GCN) models coupled with Long Short-Term Memory (LSTM) networks [35].

Graph Neural Networks are a class of deep learning models specifically designed to handle graph-structured data, which are highly relevant for analyzing complex relationships in data such as EEG recordings in tinnitus therapy. By leveraging the graph structure, GNNs can effectively capture and model the intricate connections between different nodes (e.g., EEG signal channels) and their attributes. In this study, we applied GNNs to transform EEG data into graph representations, enabling the extraction of meaningful patterns and improving the accuracy of our tinnitus therapy analysis. This novel approach combines graph representation and deep learning, achieving an impressive accuracy of 99.41%, as illustrated in Figure 2. Through the use of GNNs, our method provides a powerful tool for analyzing EEG data, offering new insights into the efficacy of various tinnitus therapies.

The utilization of GCN-LSTM models on EEG signal graphs was aimed at capturing the intricate relationships and temporal dependencies inherent to the data. These models are designed to process data structured in graphs and excel at learning patterns and representations from interconnected nodes. The LSTM component further empowers the model to grasp the long-term dependencies and temporal dynamics of tinnitus signals.

By merging graph representation and deep learning techniques, our study introduces an innovative method for scrutinizing tinnitus therapy data. This approach capitalizes on the inherent structural and temporal characteristics of EEG signals. We anticipate that this methodology will not only deepen our understanding of tinnitus but also contribute significantly to the development of more effective treatment strategies for individuals with this condition.

## 2. Materials and Methods

Here, we present a novel method for predicting tinnitus using a graph-based approach with EEG signal representations. The innovation lies in transforming the various channels of EEG signals into a novel graph network representation. This method ensures that all original information from the EEG data is preserved, enabling a comprehensive analysis that maintains the integrity of the multi-channel signal inputs. The dataset employed in this study was sourced from a publicly available benchmark. Initially, features were extracted from the EEG signals, followed by the construction of a graph based on these features. Subsequently, a model based on Graph Neural Networks was applied to acquire the representations of tinnitus. The entire tinnitus detection process is shown in Figure 3.

### 2.1. Description of Datasets

Utilizing EEG data in tinnitus research provides valuable insights into the underlying brain activity associated with this condition. EEG is a non-invasive technique that records the electrical activity of the brain through electrodes on the scalp. Analysis of EEG signals enables researchers to observe and study the neural dynamics and patterns associated with tinnitus.

#### Acoustic Therapies for Tinnitus Treatment: An EEG Database

The EEG Database for Acoustic Therapies for Tinnitus Treatment was created by researchers at the Tecnológico de Monterrey in Mexico [21,36]. The dataset contained EEG recordings of 103 participants who were treated with five different acoustic therapies for tinnitus: relaxing music, tinnitus retraining therapy (TRT), therapy for enriched acoustic environment (TEAE), binaural beats therapy (TBB), and auditory discrimination therapy (ADT). The EEG recordings were taken under four different conditions: rest, listening to the acoustic therapy, auditory stimulation based on acoustic therapy (passive mode), and identification of common auditory stimuli, for example, cell phone ringing (active mode).

In this dataset, all five acoustic therapies were effective in reducing tinnitus perception, but TRT was the most effective therapy. The researchers also found that the effects of acoustic therapies were more pronounced in the passive mode than in the active mode.

Figure 4 shows the process of administering acoustic therapy for one hour per week while monitoring the EEG signals. During the EEG session, the patients engaged in various activities. Additionally, the control group consisted of healthy participants without tinnitus. The other group comprised patients who experienced tinnitus.

This study included both a control group and an auditory discrimination therapy (ADT) group to investigate the effects of acoustic therapies on tinnitus perception and neuroplastic changes. The control group comprised healthy volunteers who did not experience tinnitus. These individuals are an essential comparison group for evaluating the specific impact of ADT therapy on patients with tinnitus. The control group followed the same experimental protocol as the ADT group, which included the administration of acoustic therapy and participation in EEG monitoring sessions. The ADT group comprised patients with tinnitus who underwent specialized therapy aimed at improving their auditory discrimination ability. This therapy involves exposing patients to different sounds and training them to differentiate between auditory patterns or frequencies. ADT aims to reduce tinnitus perception by enhancing the ability to discriminate auditory stimuli. The EEG data of the ADT group provide insights into the neuroplastic changes associated with this therapy, shedding light on the mechanisms underlying the effectiveness of the therapy.

### 2.2. Preprocessing

In this study, we performed preprocessing and analysis of EEG data from two distinct groups: auditory discrimination training (ADT) and control. The EEG data underwent several processing steps as described below.

#### 2.2.1. Data Loading and Filtering

This study employs Python for signal processing purposes. Utilizing the SciPy library, specifically its signal processing functions, the Butterworth-type bandpass filter is utilized to eliminate frequencies outside a designated range.

The initial step involves loading the EEG data and subsequently applying a Butterworth-type bandpass filter to suppress frequencies that fall outside the desired range. The transfer function $H\left(s\right)$ of the Butterworth filter [37] is shown in
(1)$$H\left(s\right)=\frac{\left(s^{n}\right)}{\left(s^{n}\right)+\left(\omega c^{n}\right)}$$
where *H*(*s*) is the transfer function of the filter, *s* is the complex frequency variable, *n* is the filter order, and ωc is the cutoff frequency. Our study employs Python’s SciPy library to both design and implement a Butterworth bandpass filter with a filter order of 4, and cutoff frequencies within the conventional EEG range 0.5–1 Hz (to eliminate low-frequency drift) and 40–50 Hz (to eliminate high-frequency noise).

The frequency response $H\left(f\right)$ of the Butterworth filter is expressed in Equation (2) as follows:(2)$$H\left(f\right)=\frac{1}{\sqrt{1+{\left(\frac{f}{\omega c}\right)}^{2n}}}$$
where $H\left(f\right)$ denotes the frequency response of the filter, *f* represents the frequency, *n* indicates the filter order, and ωc is the cutoff frequency.

Using these equations, the mathematical representation of the Butterworth bandpass filter can be expressed by Equation (3) as:(3)$$f\left(t\right)=\int \left[f\left(EEG\left(t\right)\right)\cdot h\left(t\right)\right]dt$$
where $f\left(t\right)$ denotes the filtered EEG signal at time t, $f\left(EEG\left(t\right)\right)$ represents the input EEG signal after passing through the Butterworth filter, $h\left(t\right)$ is the impulse response of the filter, and the integral signifies the convolution operation between the filtered signal and impulse response. Figure 5 depicts the application of a filter to EEG signals. The blue curve represents the original data, whereas the red curve shows the data after the filtering process, indicating the cleaned or processed signals.

The blue curve in the figure represents the raw EEG signal data. These data are typically collected from electrodes placed on the scalp and contain various frequencies, including noise and artifacts that can obscure the true signal of interest. The red curve shows the EEG signal after it has undergone a filtering process. This process aims to remove unwanted noise and artifacts from the original signal, leaving a cleaner and more interpretable set of data. The filtered signal retains the essential components of the EEG while minimizing distortions and extraneous information.

Filtering is a crucial step in EEG signal processing, as it serves multiple essential functions. Firstly, it reduces noise from various sources like power line interference, muscle activity, and movement artifacts, thereby enhancing the overall quality of the signal. Secondly, filtering attenuates significant artifacts such as eye blinks and heartbeats, which can obscure brain activity in EEG recordings. Lastly, it allows for the isolation of specific frequency bands, facilitating the focused analysis of different brain states and activities.

The filtered data, depicted by the red curve, should maintain the integrity of the EEG signal by preserving essential features like peaks and troughs that correspond to neural activity. Moreover, the filtering process should enhance signal clarity, removing extraneous noise and artifacts to provide a clearer picture of the underlying brain activity, thus aiding in the identification of patterns and anomalies.

#### 2.2.2. Bad Channel Removal

To enhance the data quality, channels that exhibit characteristics such as flatness, high-frequency noise, or low correlation with neighboring channels are identified as bad channels and subsequently removed. This process entails evaluating various metrics to assess the quality of each channel and eliminating those that do not meet the specified criteria.

#### 2.2.3. Event Definition

Events were defined based on the markers found in the data, enabling the identification of specific time intervals of interest. This step involves identifying particular time points or intervals within the EEG data that correspond to specific events or stimuli.

#### 2.2.4. Windows Creation

Continuous EEG data were divided into smaller time windows, known as epochs, with predefined start and end times for each epoch. This segmentation allows for the examination of distinct temporal segments within the EEG data.

#### 2.2.5. Epoch Cleaning

The Random Sample Consensus (RANSAC) algorithm was utilized to detect and eliminate artifacts within epochs, thereby enhancing the signal-to-noise ratio. The RANSAC algorithm iteratively identifies and removes outliers or artifacts within each epoch. The motivation for using the Random Sample Consensus (RANSAC) algorithm for motion artifact elimination in our work was based on its robustness to outliers, flexibility, and adaptability to various types of noise, which are common in EEG signals. RANSAC’s iterative approach effectively fits a model to inlier data while ignoring outliers, making it particularly suitable for our application. Additionally, RANSAC has a proven track record in noisy data applications such as computer vision, and comparative studies showed it outperformed other well-established methods like ICA and Wavelet Transform in maintaining EEG signal integrity. Its computational efficiency also supports real-time EEG analysis, crucial for applications such as brain–computer interfaces. Integrating RANSAC with Graph Convolutional Networks (GCNs) further enhances our approach, as GCNs benefit from the clean, artifact-free data RANSAC provides, enabling a more accurate and reliable analysis of EEG signals.

#### 2.2.6. Power Spectral Density (PSD) Computation

The frequency bands were derived by computing the power spectral density (PSD) of the EEG signal, which provides insights into the power distribution across different frequencies. The $PSD\left(f\right)$ [38] is determined as
(4)$$PSD\left(f\right)={\left|X\left(f\right)\right|}^{2}$$

#### 2.2.7. Average PSD Calculation

To obtain a representative measure of the spectral content for each epoch, the average PSD values across channels were computed.

### 2.3. Features Extractions

After preprocessing the EEG signals, various features were extracted from the EEG signals. These features provide insights into different aspects of the signal, including the time-domain, frequency-domain, and time-frequency characteristics. The extracted features were used to analyze the differences between the tinnitus and control groups. The feature-extraction process involves several steps, as described below.

First, the EEG signals were normalized. This step ensures that the signal data have zero mean and unit variance, which is important for accurate feature extraction.

Next, we computed various time-domain features from the normalized signals. These features include the mean, standard deviation, root mean square (RMS), and variance. The mean represents the average value of the signal, whereas the standard deviation indicates the spread of the signal values around the mean. The RMS provides a measure of the overall magnitude of the signal, and the variance quantifies the variability of the signal.

Moving to the frequency domain, we estimated the power spectral density (PSD) of the signal. The PSD represents the distribution of power across different frequencies. From the PSD, we extracted the maximum frequency, which corresponded to the frequency with the highest power. In addition, we computed the mean frequency by averaging the product of the frequencies and their corresponding power values. The bandwidth was calculated by numerically integrating PSD. Figure 6 represents a plot of different PSDs from different bands on the dataset.

To capture time-frequency information, we employed the Piecewise Aggregate Approximation (PAA) technique. The PAA algorithm divides the signal into smaller segments and extracts the PAA-transformed signals. Then, we computed, which reveals the energy distribution of the signal across both time and frequency. From the spectrogram, we derived the maximum time, which represents the time at which maximum energy occurs. Similarly, the mean time frequency is computed as the weighted average of the time values and their corresponding energies. We also obtained the maximum frequency at each time point and calculated the mean frequency over time.

In addition to the above features, we included statistical features such as signal statistics and average amplitude changes. These features provide further insights into the characteristics and dynamics of the signal.

In total, the extracted features provide a comprehensive representation of the EEG signal, encompassing temporal, spectral, and temporal–spectral aspects. By analyzing these features, we aim to identify differences between the healthy and tinnitus EEGs, which can provide valuable insights into the effects of auditory discrimination training on EEG patterns.

While it is true that EEG signals are highly non-linear and non-stationary, our approach leverages the power of graph representation to effectively capture the underlying structure and dynamics of these signals. By representing EEG channels as nodes and their interactions as edges, our graph method captures spatial and temporal relationships, providing a rich structural context that enhances feature reliability. This structure, combined with GCNs, allows for the learning of intricate patterns and non-linear interactions within the data. GCNs process the entire network of features simultaneously, improving the model’s ability to handle the complexities of EEG signals. Additionally, the graph representation’s robustness to noise and artifacts further ensures the reliability of the features. Our extensive experiments have validated that even basic features, when integrated into this graph framework, lead to superior performance in EEG analysis tasks. Thus, the graph-based approach, enhanced by GCNs, provides a robust, scalable, and effective method for capturing the complex characteristics of EEG signals.

### 2.4. Graph Representation

To apply the GNN-LSTM model, a graph was constructed to capture the relationships between the feature vectors in both the healthy and tinnitus EEG groups. The graph construction process involves connecting nodes based on the Euclidean distance between the feature vectors. For each feature type associated with a channel, a node was added to the graph. Subsequently, edges were created between pairs of nodes if the Euclidean distance between their corresponding feature vectors was below the threshold of 0.2. This connectivity criterion ensures that only closely related feature vectors are linked in the graph. The resulting graph served as the basis for analyzing the interdependencies between different feature types. The selection of this threshold value to define connections between nodes in our graph representation was a critical aspect of our methodology. We determined this threshold through careful consideration of the underlying EEG data properties and the desired level of connectivity to capture relevant information while minimizing noise. Our approach aimed to strike a balance between capturing meaningful connections and reducing spurious ones, thereby enhancing the interpretability and reliability of our graph representation.

The graph structure was constructed to facilitate the application of a GNN-based GCN model for classification tasks. The construction of the graph aimed to establish a graph-node architecture, which is a prerequisite for GNN-based models.

#### 2.4.1. Graph Initialization

An empty undirected graph, denoted as G, was created.

#### 2.4.2. Node Addition

Each feature vector in the dataset, represented by $X_{i}$, was associated with a unique node in the graph. The number of nodes in the graph corresponded to the number of samples in the dataset.

#### 2.4.3. Edge Creation

Pairwise comparisons between nodes were performed to compute the Euclidean distance, dist${(X}_{i},\;X_{j})$, between their respective feature vectors.

#### 2.4.4. Edge Addition

An edge is introduced between nodes *i* and *j* if the Euclidean distance between their feature vectors is less than a predetermined threshold value, *ε*. This criterion ensures that only the nodes with similar feature vectors are connected in the graph.

The resulting graph G represents the relationships between the feature vectors, with nodes representing the samples and edges denoting their connections. This graph structure enables the utilization of the GCN model for classification tasks.

By exploiting the interconnectedness inherent in the feature vectors represented by the graph, the GCN model demonstrated the ability to effectively capture intricate patterns and dependencies, thereby enhancing classification accuracy and overall performance.

Algorithm 1 outlines the process of converting EEG features into graph nodes and edges. The algorithm is initiated by creating an empty graph, G. It then iterates through the rows of feature matrix X, introducing a node for each sample to G. Subsequently, it traverses all pairs of nodes in G, excluding self-loops. For each node pair, the algorithm calculated the Euclidean distance between the corresponding feature vectors. An edge is established between nodes if the distance falls below the specified threshold. Ultimately, the algorithm produces the constructed graph, G, depicted in Figure 7, illustrating a comparison of extracted features from two signal types: non-tinnitus and tinnitus signals. The plot distinctly shows that nodes representing tinnitus signals, marked in red, typically exhibit slightly higher values compared to nodes representing non-tinnitus signals, marked in blue. This systematic procedure guarantees the efficient conversion of EEG features into a graph structure, facilitating subsequent analysis using the GCN model.
**Algorithm 1**. Algorithm to convert EEG features set to graph nodes and edges.INPUT: X, *y*OUTPUT: G (graph structure)**Initialize** an empty graph GFor i=0 to i < number of rows in X, **do**:**Add** the *i*-th row as a node in the graph GFor i=0 to i < length of X, **do**:    For j=i+1 to j < length of X, **do**:      i. **If**
*i* == *j*, **then** continueii. Calculate the Euclidean Distance between X(i,:) and X(j,:)  and **store** it in dist     iii. If d<0.2 and d(X(i,:), X(j,:)), **then** add an edge between nodes i and j in G**Return** G

### 2.5. The Proposed GCN-LSTM Model

The model proposed in this study was designed to combine a GCN and LSTM for a specific task. The GCN-LSTM model aims to leverage the strengths of both the GCN and LSTM to effectively analyze graph-structured data and capture temporal dependencies.

The GCN-LSTM model consists of three main components: GCN, LSTM, and their integration. The GCN component is responsible for processing the graph-structured data and extracting meaningful representations. It consists of multiple GCN layers (GCNConv) that operate on input data with increasing levels of abstraction. Each GCN layer takes the input data and edge indices of the graph as inputs, applies graph convolutional operations, and utilizes the ReLU activation function to introduce nonlinearity. The output of the final GCN layer is passed through an ReLU activation function to obtain the final graph representation.

The GCN component consisted of multiple GCN layers. We denote the input to the GCN component as $X^{0}$, which represents the initial node features. Each GCN layer applies the following operations for *l* = 1 to L, where L is the total number of GCN layers.
(5)$$X^{l}=ReLU\left(D_{hat}^{-\frac{1}{2}}*A_{hat}*D_{hat}^{-\frac{1}{2}}*X^{l-1}*W^{l}\right)$$
where $A_{hat}$ is the adjacency matrix of the graph augmented with self-connections ($A_{hat}=A+I$), and *I* is the identity matrix. $D_{hat}$ is the diagonal matrix of the node degrees in $A_{hat}$ ($D_{{hat}_{ii}}=\sum A_{{hat}_{ij}}$). $X^{l-1}$ is the input node feature in the (l−1)th layer. $W^{l}$ denotes the weight matrix of the lth layer.

After the GCN layers, the output $X^{l}$ from the last GCN layer was reshaped to a time sequence format to be fed into the LSTM layer. The reshaping operation converts the fixed-size representation of the nodes into a time-sequence representation. We denote the reshaped input as *X_*reshaped.

The LSTM component captures the temporal dependencies in the graph representations obtained from the GCN. It takes the output of the final GCN layer as the input and applies the LSTM architecture, which consists of an LSTM layer followed by a fully connected (linear) layer. The LSTM layer processes the input sequence across time steps and captures the sequential information. The hidden state of the LSTM at the last time step was then fed into the fully connected layer to obtain the final output.

The LSTM layer processes the input *X_*reshaped across time steps. We denote the input to the LSTM layer at time step t as $X_{LSTM}^{t}$. The LSTM layer applies the following operations.
(6)$$H^{t},\left(h_{n},c_{n}\right)=LSTM\left(X_{LSTM}^{t},h_{n-1},c_{n-1}\right)$$
where $H^{t}$ represents the hidden state output of the LSTM at time step *t*. $h_{n}$ and $c_{n}$ are the hidden and cell states of the LSTM, respectively, at the nth time step.

The output of the LSTM layer was then fed into a fully connected layer to obtain the final predictions or classifications.
(7)$$Output=FC\left(H^{T}\right)$$
where *T* is the total number of time steps.

The GCN-LSTM model integrates the GCN and LSTM components by sequentially applying them. The graph representation obtained from the GCN was reshaped and passed as input to the LSTM. LSTM processes the reshaped input across a specified number of time steps, capturing temporal dependencies. The final output of LSTM represents the predictions or classifications for a given task.

The parameters of the GCN layers, including the weight matrices $W^{l}$, were learned during the training process to minimize the loss function. Similarly, the parameters of the LSTM layer, including the weight matrices and biases, were also learned through backpropagation and gradient descent.

In our study, we employ PyTorch to implement both GCN and LSTM models. To mitigate data leakage and overfitting, we adopt several strategies during the training and testing phases. Specifically, we use ten-fold cross-validation, early stopping, and regularization to enhance model generalization and prevent overfitting. Additionally, we carefully preprocess the data, partitioning it into distinct training, validation, and testing sets to ensure the models are evaluated on independent datasets.

## 3. Results and Discussion

The GNN LSTM model was trained using a graph constructed from the features of the dataset. The training process focused on optimizing the model parameters to minimize the loss function and enhance its capability to detect tinnitus. Following 1500 epochs of training, the model demonstrated promising results. The loss function yielded a value of 0.2063, indicating the extent of error between predicted and actual values. Accuracy, standing impressively high at 0.9941, denotes the proportion of correct predictions out of the total predictions made. Recall, at 0.9741, signifies the ability of the model to correctly identify positive instances from all actual positives. Precision, scoring a perfect 1.0000, represents the model’s capability to correctly identify positive predictions from all positive instances it predicted. The F1 measure, a harmonic mean of precision and recall, reached 0.9700, providing a balanced assessment of the model’s performance. Sensitivity, at 0.9841, reflects the proportion of true positive instances correctly identified by the model. Kappa Cohen’s coefficient, indicating the agreement between predicted and actual values while accounting for chance, scored notably high at 0.9806. Lastly, the area under the curve (AUC) value, measuring the model’s ability to distinguish between classes, was recorded at 0.991, suggesting strong discriminatory power. These metrics collectively reflect the robustness of the model in effectively identifying tinnitus in the dataset.

The results obtained from our study showed the efficacy of the GNN LSTM model in accurately detecting tinnitus using the provided dataset. The model achieved a remarkably high accuracy of 0.9941, as illustrated in the accuracy graph in Figure 8. This exceptional accuracy underscores the ability of the model to effectively distinguish tinnitus within the analyzed data.

The scatter plot, as depicted in Figure 9, and results from the application of principal component analysis (PCA) on the output of our trained tinnitus detection model reveal discernible clusters of dots. These clusters indicate the potential effectiveness of the model in distinguishing individuals with and without tinnitus, particularly in response to various therapeutic interventions. The evident separation of dots implies that the model acquired discriminative features, enabling it to classify individuals based on the presence or absence of tinnitus.

This observation underscores the model’s proficiency in capturing the underlying patterns or characteristics associated with tinnitus with distinct groups likely representing different therapeutic outcomes. This distinct separation further suggests that the model’s learned representations are robust and generalize effectively to unseen instances, emphasizing its practical utility in the classification of tinnitus. The insights derived from the scatter plot underscore the potential of our model as a valuable tool for evaluating tinnitus treatment outcomes and advancing our understanding of this condition.

The confusion matrix for our tinnitus detection model highlights its performance metrics. It comprises four key values: true positives (0.97), false positives (0.03), false negatives (0.02), and true negatives (0.98). These values indicate the accuracy of the model in correctly identifying individuals with and without tinnitus, demonstrating high precision and reliability.

The confusion matrix is an essential tool for evaluating the performance of our tinnitus detection model, providing a clear summary of prediction results on a classification problem. The matrix consists of four key components:

True Positives (TPs): These are cases where the model correctly identifies individuals who have tinnitus. In our model, the true positive rate is 0.97, meaning that 97% of the individuals who actually have tinnitus were correctly identified by the model.

False Positives (FPs): These represent cases where the model incorrectly identifies individuals as having tinnitus when they do not. Our model has a false positive rate of 0.03, indicating that 3% of the individuals who do not have tinnitus were incorrectly classified as having the condition.

False Negatives (FNs): These are cases where the model fails to identify individuals who actually have tinnitus. The false negative rate for our model is 0.02, which shows that 2% of the individuals with tinnitus were not detected by the model.

True Negatives (TNs): These represent cases where the model correctly identifies individuals who do not have tinnitus. The true negative rate is 0.98, indicating that 98% of the individuals without tinnitus were correctly identified.

These values together demonstrate the high precision and reliability of our tinnitus detection model, with a strong ability to correctly identify both positive and negative cases. This comprehensive evaluation highlights the model’s effectiveness in distinguishing between individuals with and without tinnitus.

The high true positive and true negative values presented in Figure 10 indicated the effectiveness of the model in accurately classifying individuals with and without tinnitus, respectively. Furthermore, the low values for false positives and false negatives suggest minimal misclassification, underscoring the robustness of the model. In summary, the outcomes depicted in the confusion matrix emphasize the promising performance of our tinnitus detection model and its potential as a valuable tool for precisely identifying the presence or absence of tinnitus.

Figure 11 provides compelling evidence that tinnitus retraining therapy (TRT) is more effective compared to alternative tinnitus treatments. This increased efficacy is attributed to TRT’s holistic approach, which focuses on habituation, helping the brain reclassify tinnitus as a neutral signal. TRT combines sound therapy to mask tinnitus and expedite habituation, counseling sessions to alleviate the negative emotional impact, and a personalized treatment plan tailored to individual needs. This comprehensive strategy, supported by an expanding body of clinical evidence, positions TRT as a highly promising treatment option for many tinnitus sufferers, offering hope for improved outcomes and quality of life. Moreover, by examining patient recovery data across diverse therapies, we generated a graph that visually represents the efficacy of TRT in comparison to alternative approaches. This graph depicts recovery percentages or reductions in tinnitus severity scores over time.

The confusion matrix shows the performance evaluation of the GCN-LSTM model on the tinnitus dataset, likely reporting metrics such as accuracy, precision, recall, and F1 score, which are standard for evaluating classification models. Figure 11, on the other hand, appears to illustrate the effectiveness of different tinnitus therapies, presumably based on patient recovery data or reductions in tinnitus severity scores over time. The GCN-LSTM model evaluated may have been used to classify patients’ responses to various tinnitus therapies, with the resulting classification accuracies contributing to the assessment of therapy effectiveness shown in Figure 11. The patient recovery data or reductions in tinnitus severity scores mentioned in Figure 11 might have been used as ground truth labels for training and evaluating the GCN-LSTM model, whose performance is depicted in the confusion matrix. By leveraging the tinnitus detection capabilities of the GCN-LSTM model, as evaluated through the confusion matrix, we can derive the recovery rate of patients from tinnitus therapies. If the model did not detect tinnitus in a patient after therapy, it implies that the therapy was effective in alleviating or reducing the patient’s tinnitus perception. Thus, the model’s accuracy in correctly classifying the absence of tinnitus (true negatives in the confusion matrix) can be linked to the recovery rate or effectiveness of the tinnitus therapies shown in Figure 11. This connection between the confusion matrix and Figure 11 indicates that the model’s performance in identifying the absence of tinnitus post-therapy reflects the therapy’s effectiveness, as represented by the recovery percentages or reductions in tinnitus severity scores in Figure 11.

In comparison, Alonso-Valerdi et al. [19] evaluate the psycho and neurophysiological effects of various therapies, including TRT, neuro-modulation techniques (TEAE and ADT), and distress relief methods (BBT and music therapy), on a sample of 60 tinnitus sufferers and 11 control participants. They find that neuro-modulation therapies effectively reduced stress and anxiety without side effects, while TRT and music therapy primarily alleviated anxiety. However, TRT and BBT were also associated with increased anxiety. The researchers suggest that a more reliable method is needed to assign therapies based on patient profiles.

Another study [39] confirms TRT’s effectiveness in reducing tinnitus and stress but cautioned against its use for patients with pre-existing anxiety. It finds BBT, TEAE, and ADT to be similarly effective in reducing tinnitus perception and managing stress and anxiety, with ADT presenting fewer side effects and TEAE not exacerbating tinnitus perception. Conversely, music therapy was deemed less effective as it could potentially worsen tinnitus perception. The authors emphasize the importance of tailoring therapies to individual patient conditions and recommend developing applications to monitor and register the daily use of acoustic therapies.

Overall, while all three studies underscore TRT’s efficacy, they also highlight the need for personalized treatment tailored to individual patient profiles. These studies offer varied insights into the neurophysiological and psychological impacts of different acoustic therapies. Additionally, this study presents data showing that TRT yields superior recovery rates and more significant reductions in tinnitus severity compared to other therapies, reinforcing its potential as a highly effective treatment option.

## 4. Conclusions

In this study, pertinent features were extracted using a tailored algorithm to construct a graphical representation, with nodes connected based on a specified distance threshold. The subsequent application of the GNN_LSTM model, which combines Graph Neural Networks and Long Short-Term Memory networks, demonstrated remarkable efficacy in accurately identifying tinnitus cases, showcasing a notable accuracy rate of 99.41%. This amalgamation of graph representation and deep learning techniques introduces a novel methodology for scrutinizing tinnitus therapy data, offering advanced insights into the neural patterns associated with this condition. The success of our approach not only underscores its potential for advancing more efficacious treatment modalities but also has significant implications for tinnitus diagnosis and treatment. These findings pave the way for improved diagnostic tools and personalized treatment strategies. Moving forward, future research directions include refining graph construction algorithms, integrating multimodal data for comprehensive understanding, exploring real-time monitoring and intervention possibilities, conducting large-scale clinical validations, and developing patient-specific treatment recommendations. These efforts aim to enhance the accuracy, generalizability, and practical utility of the model in diverse clinical settings, ultimately contributing to advancements in tinnitus research and patient care.

## Figures and Tables

**Figure 1 biomedicines-12-01404-f001:**
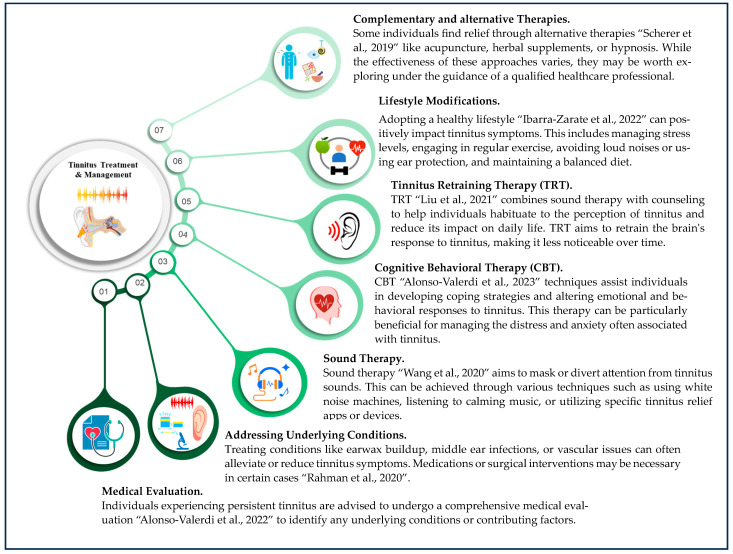
Overview of prevalent strategies utilized for tinnitus management across all age demographics [3,19,20,21,22,23,24].

**Figure 2 biomedicines-12-01404-f002:**
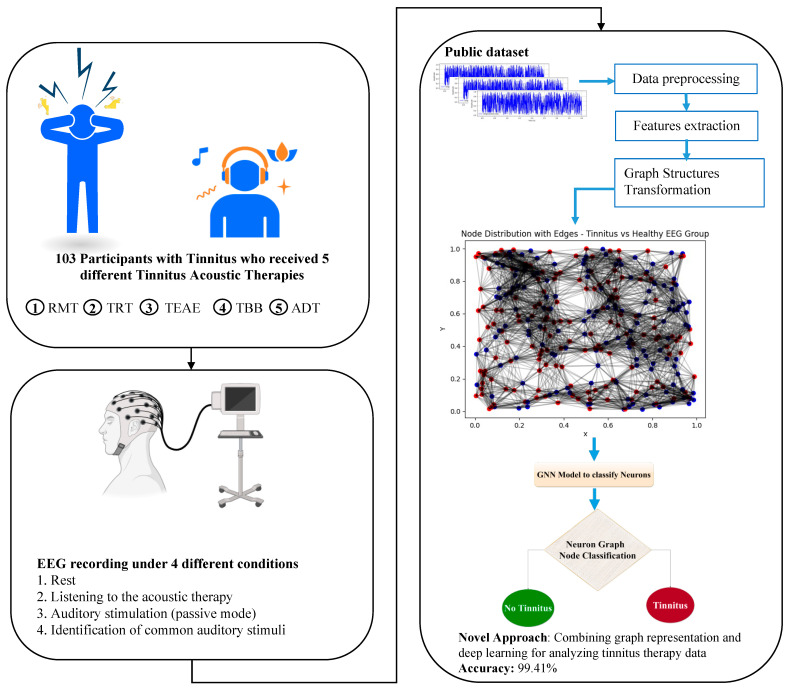
Graph-based EEG analysis in tinnitus therapy. RMT: relaxing music therapy; TRT: tinnitus retraining therapy; TEAE: therapy for enriched acoustic environment; TBB: binaural beats therapy; ADT: auditory discrimination therapy.

**Figure 3 biomedicines-12-01404-f003:**
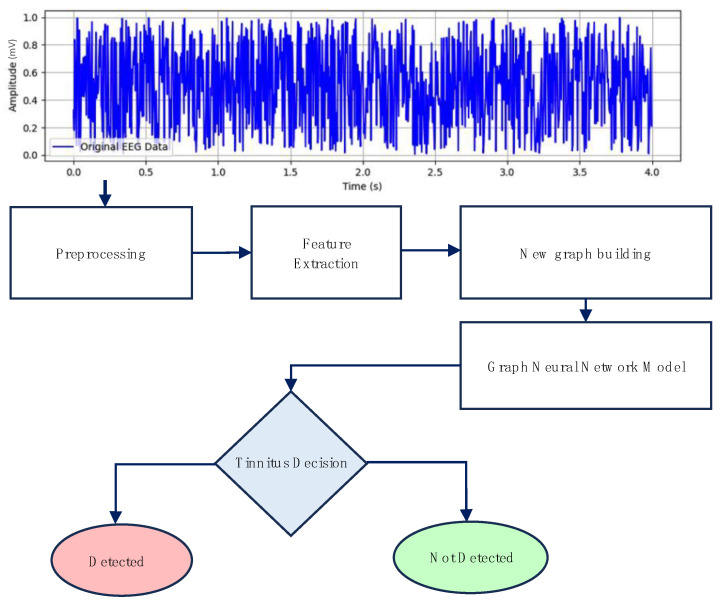
Schematic depiction of the proposed tinnitus detection model architecture utilizing GNN applied to tinnitus EEG signals.

**Figure 4 biomedicines-12-01404-f004:**
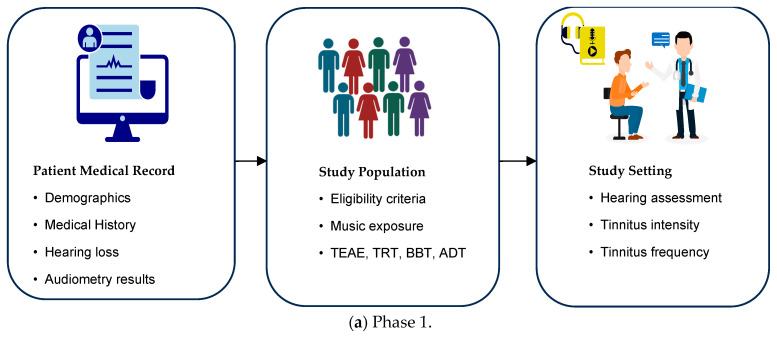
Procedure of administering weekly one-hour acoustic therapy (phase 1) with concurrent monitoring of EEG signals (phase 2) [21].

**Figure 5 biomedicines-12-01404-f005:**
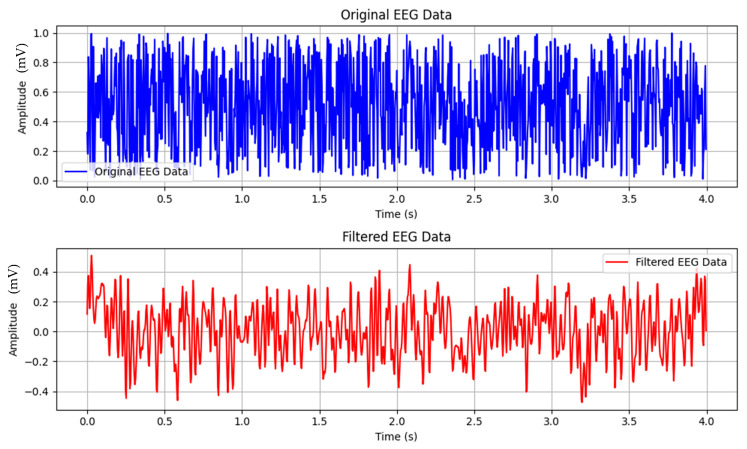
Representation of the application of the filter to EEG signals. The blue curve corresponds to the original data, whereas the red curve indicates the data post-filtering, highlighting the cleaned or processed signals.

**Figure 6 biomedicines-12-01404-f006:**
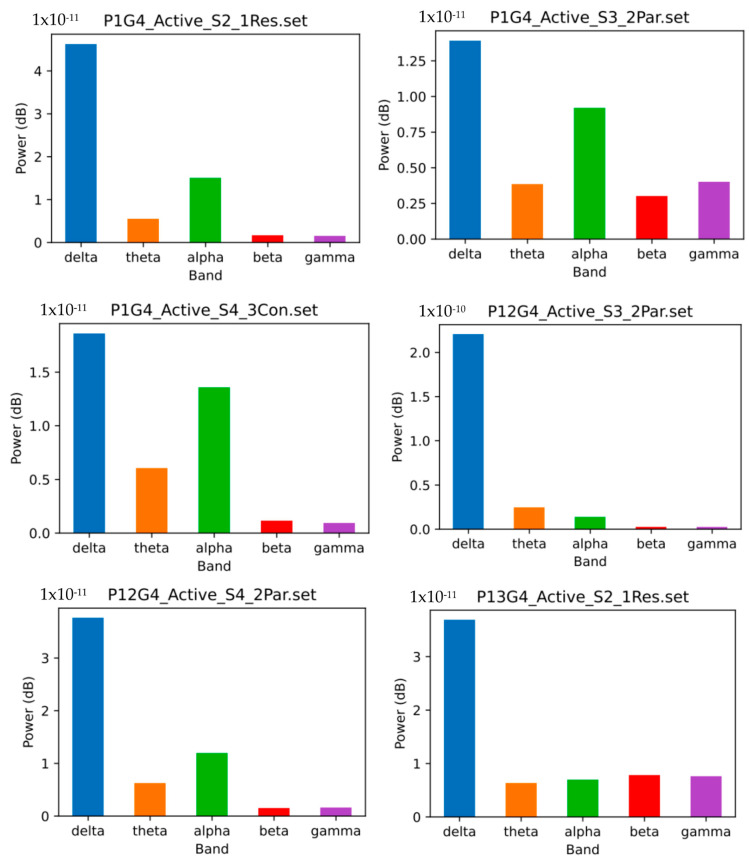
Plot illustrating diverse power spectral density (PSD) values across various frequency bands within the dataset.

**Figure 7 biomedicines-12-01404-f007:**
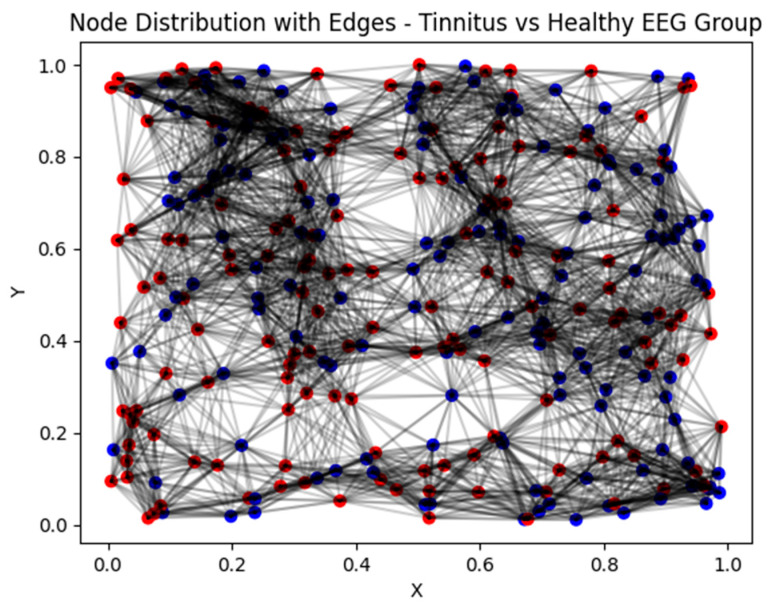
Graph G visualizing relationships among feature vectors, with nodes representing individual samples and edges denoting their connections. The tinnitus EEG group is represented by red nodes, while the healthy EEG group is represented by blue nodes.

**Figure 8 biomedicines-12-01404-f008:**
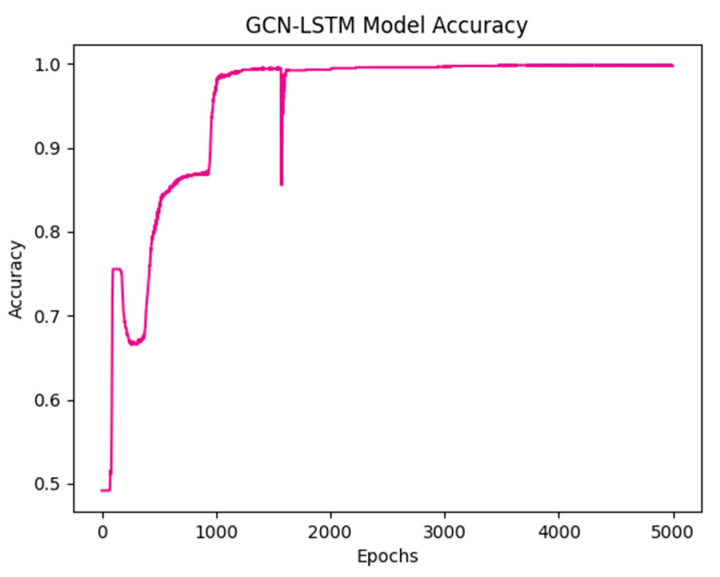
Learning curve on identifying seizure state.

**Figure 9 biomedicines-12-01404-f009:**
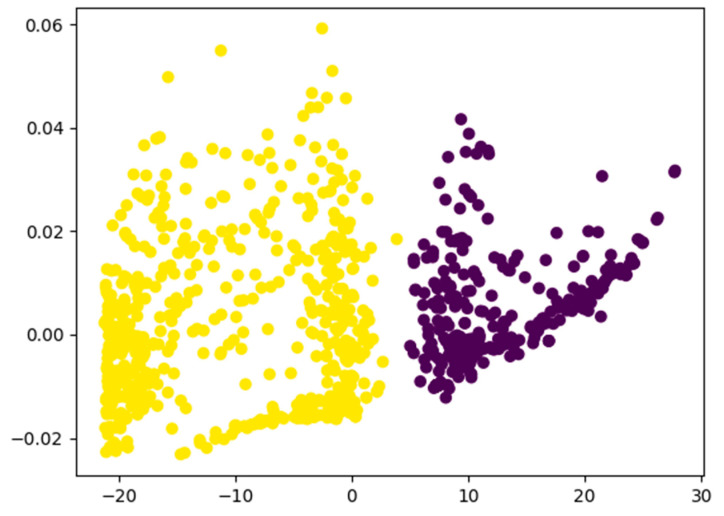
Scatter plot derived from the application of principal component analysis (PCA) on the output of our trained tinnitus detection model. Yellow dots represent the true class, while purple represent the false class.

**Figure 10 biomedicines-12-01404-f010:**
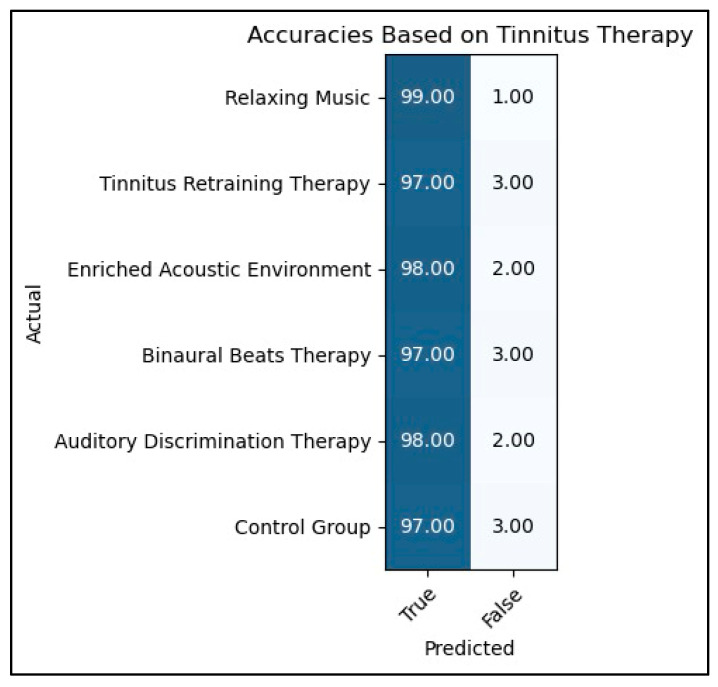
Effective classification of individuals with and without tinnitus.

**Figure 11 biomedicines-12-01404-f011:**
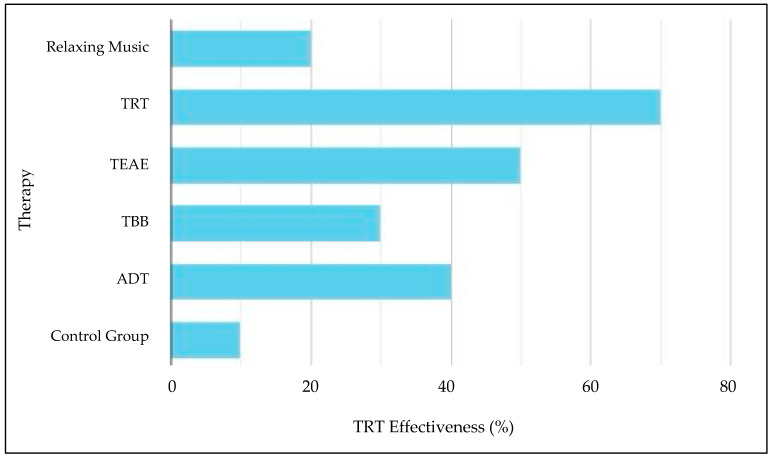
Effectiveness of different therapies of tinnitus.

## Data Availability

The datasets are publicly available from the links https://doi.org/10.17632/kj443jc4yc.1, accessed on 13 September 2023.

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
