# Peer review of "Graph-Based Electroencephalography Analysis in Tinnitus Therapy"

_biomedicines, 2024, doi:10.3390/biomedicines12071404_

Round 1

Reviewer 1 Report

Comments and Suggestions for Authors

The article entitled “Graph-Based EEG Analysis in Tinnitus Therapy” is well-written and, from my point of view, would be of interest for the readers of Biomedicines. In spite of this and before its publication, I would recommend authors to perform the following changes that from my point of view would serve for the manuscript improvement.

First of all, the article should include a more in-depth presentation of GNN.

Please check and review Figure 3 thinking if it would be possible to express the same with a more simple figure.

Line 298, please introduce any bibliographical reference to the  Butterworth filter.

Figure 4 should be explained more in-depth in the text.

Line 352-353: a reference about PSD is required.

Figure 5: figures must be individually labelled and also enlarged as they are dificult to read.

Metrics from line 518 to 526 should be presented in a paragraph explaining them instead of as a list.

Figure 9 with the confusion matrix is not required. It can be explained either as a paragraph or in a Table.

Author Response

Dear Reviewer,

Please find the answers to your comments. We have carefully addressed all the issues you pointed out and made corresponding modifications to the manuscript. The updated text is highlighted in yellow.

We would like to thank you for your constructive and relevant comments, as well as for your suggestions.

Best regards.

The authors.

Reviewer 2 Report

Comments and Suggestions for Authors

This paper introduces the GCN-LSTM approach to analyze tinnitus therapy using EEG. The topic is interesting, and the literature is well explained. However, the results need to be adequately explained and presented. The following are my comments:

1) The abstract is too generic. The objective of this study should be clearly mentioned. For example, "this study focuses on analyzing the dataset related to tinnitus theraph..." What is the motivation to analyze such type of data? The author's objective is to propose which therapy method is better to treat this issue. The abstract should be clear and contain objectives, methods, results, and applications.

2) In materials and Methods, the authors mentioned "Here, we present a novel method for predicting tinnitus using a graph-based approach with EEG signal representations." What is a novel? The graph-based method are well established and widely applied on EEG data. Applying this technique to EEG data for tinnitus is not a novel idea. The use of this word should be omitted from the text. Most of the research is just the application of different techniques; there is no novelty in applying already established methods. 

3) Figure 1: Why are there two graph blocks? There is a typo in the preprocessing block. What do you mean by feature? This figure is very basic, and these steps are known to everyone. I suggest modifying this figure and replacing it with more details on including the method in each block.

4) Add a reference for "The EEG Database for Acoustic Therapies for Tinnitus Treatment was created by researchers at the Tecnológico de Monterrey in Mexico."

5) Figure 3: Text and block border are overlapping. Also, add a reference here too.

6) Data loading and filtering. Which platform or software was used in this study? Include filter order and cutoff frequencies used for filtering.

7) What kind of filtering was performed on this data? Why is there a drift at the data's start and end? What is the objective of this figure? Are the authors proposing some filtering method? What is the unit on the y-axis?

8) What was the motivation for using the "Random Sample Consensus (RANSAC) algorithm" for the motion algorithm? There were other well-established methods to eliminate artifacts. 

9) The features extracted were very basic. EEG signals are highly non-linear and non-stationary. What are the reliability of these features? 

10) Graph representation: how was the threshold value selected? Was the ablation study done to choose this value?

11) Figure 6: what are blue and red nodes?

12) Which routine was used to implement GCN and LSTM? How were data leakage and overfitting avoided? More information is needed about training and testing.

13) The quality of all figures is very poor. The presentation of all figures shall be improved. Also, the captions and labels must be revised.  

14) How confusion matrix was obtained? This is using cross-validation or leaving one out. It is better to show the number of epochs correctly classified instead of percentage.

15) What is Figure 11's significance? How does this figure relate to the results? How this figure was obtained?

16) The conclusion is superficial. No conclusion can be obtained by reading this conclusion. It should be revised by including quantitative data.

17) No comparative analyses were presented. The authors should include a comparative analysis with the studies using the same dataset to claim the supremacy of their proposed method. 

Author Response

(The authors gave the same response as above.)

Round 2

Reviewer 2 Report

Comments and Suggestions for Authors

The authors have addressed most of the comments raised in the initial version. The manuscript has been improved, but there are some minor suggestions.

i) Figure 7: Add a legend or specify blue and red dots in the caption. 

ii) Add details of cross-validation, e.g., ten-fold or five-fold.

iii) Improve the quality of Figure 11. 

Author Response

(The authors gave the same response as above.)
